# Multi-Sensors Geophysical Monitoring for Reinforced Concrete Engineering Structures: A Laboratory Test

**DOI:** 10.3390/s21165565

**Published:** 2021-08-18

**Authors:** Luigi Capozzoli, Giacomo Fornasari, Valeria Giampaolo, Gregory De Martino, Enzo Rizzo

**Affiliations:** 1Institute of Methodologies for Environmental Analysis, National Research Council, C.da S. Loja, 85050 Tito Scalo, PZ, Italy; luigi.capozzoli@imaa.cnr.it (L.C.); valeria.giampaolo@imaa.cnr.it (V.G.); gregory.demartino@imaa.cnr.it (G.D.M.); 2Department of Physics and Earth Sciences, University of Ferrara, 44121 Ferrara, Italy; giacomo.fornasari@unife.it

**Keywords:** GPR, ERT, reinforced concrete structure, laboratory test, NDT, borehole

## Abstract

Non-destructive tests are strongly required in engineering applications for monitoring civil structures. The use of compared and integrated innovative approaches based on geophysical methodologies represents an effective tool for the characterization and monitoring of reinforced concrete (RC) structures. Therefore, the main aim of the work was to improve the knowledge on the potentiality and limitations of the Ground Penetrating Radar (GPR) and the Electrical Resistivity Tomography (ERT) with electrodes disposed both on the surface and in the boreholes. The work approach was adopted on an analog model of a reinforced concrete frame built ad hoc at the Hydrogeosite Laboratory (CNR-IMAA), where simulated experiments on full-size physical models are defined. Results show the ability of an accurate use of GPR to reconstruct the rebar dispositions and detect in detail possible constructive defects, both highlighting the lack of reinforcements into the nodes and providing useful information about the safety assessment of the realized structure. The results of the ERT method defined the necessity to develop ad-hoc electrical resistivity methods to support the characterization and monitoring of buried foundation structures for civil engineering applications. Finally, the paper introduces a new approach based on the use of cross-hole ERTs (CHERTs) for the engineering structure monitoring, able to reduce the uncertainties usually affecting the indirect results.

## 1. Introduction

Non-Destructive Techniques (NDT) are widely applied in the engineering field for investigating RC elements belonging to civil structures and infrastructures. Thanks to their high resolution and repeatability, analyses based on the study of the physical property variations, it is possible to identify defects and fractures due to induced stresses, intrinsic inhomogeneities, or structural problems. Among the various NDT methods applicable for engineering issues, sonic and infrared methods represent the most used technologies for the detection of failure stage and cracking sources in engineering structures above the ground [1,2,3]. On the contrary, several indirect methods are used for ground surface investigation correlated to geotechnical engineering problems, such as Ground Penetrating Radar, geoelectrical, and seismic methods [4].

NDT focused on the detection and characterization of foundations and/or buried structures is a complex topic covering many different techniques designed to gain information about the integrity and quality of the material that makes up a foundation. Typical foundation materials are concrete, timber, steel, and rock. Foundations vary in size and shape, they are built with mixed materials and using several techniques. Therefore, to overcome some of the limitations of the surface testing techniques, a few researchers have focused on the possibilities of downhole NDT techniques [5,6].

Moreover, the integration with other methodologies represents a valid solution for mitigating the uncertainties that generally affect the results obtainable with the use of only one technique [7].

A holistic approach based on the integration of high-resolution geophysical methodologies can effectively support the characterization and analysis of engineering structures realized in masonry [8,9] or reinforced concrete [10,11]. Among the various methodologies the GPR, based on the introduction of an EM field into the material structure under investigation and the subsequent study of the scattered phenomena resulting from the EM waves propagating in the building structure [12], is one of the best methodologies to provide qualitative and quantitative information for the enhancement of knowledge about civil structures and ground conditions. Indeed, thanks to the high resolution and different investigation depth, this method is widely applied in the engineering and restoration fields for the non-invasive diagnostic of cracks, moisture analyses, constructive technology characterization (structures realized in masonry, wood, steel, etc.), and reinforcement examinations for the vulnerability and reliability analyses for reinforced concrete structures [13,14,15,16,17], for the detection of rebars, pipes, structural supports and unexpected variations of materials [18,19,20,21,22].

As regarding the analyses of foundation structures, the greater depth of investigation requested and the lack of accessibility reduce the use of GPR, therefore, innovative strategies of investigation are mandatory. One example could be the use of the electrical resistivity (ER) methods that can give a useful contribution to the investigation of the engineering structures not detectable with methods more superficial [23]. ER methods are based on Ohm’s Law and, therefore, consists of introducing current (I) into the ground surface using a pair of electrodes (commonly labeled A and B) and measuring the drop of potential (ΔV) between the second pair of electrodes (M and N) [24].

The most common ER method uses electrodes disposed on the surface, but recently a new approach was introduced, and it consists of installation them in boreholes [25] to have non-conventional configurations as regarding the electrode dispositions and sequential acquisitions. Cross-hole electrical resistivity tomography (CH-ERT) is an extension of conventional surface resistivity imaging methods. It consists in collecting apparent resistivity measurements with a plate or cylindrical electrodes in one, two, or more boreholes, placed in contact with the host soils/rocks or with the formation fluid [26]. At now, cross-hole ERTs can yield detailed information about the electrical resistivity distribution between two boreholes, with a higher resolution if compared to the conventional surface ERTs.

To date, CH-ERTs have been successfully applied in few engineering cases to investigate the resistivity distribution, to build a model of the complex subsurface geometry, such as the building foundation, and design proper remedial actions in covered karst [27]; for evaluating the safety of a ground subsidence zone in urban areas [28] and to obtain the resistivity distribution in the subsoil and reconstruct the hydraulic conductivity distribution. CH-ERTs are minimally invasive because they require the realization of small holes for installing geoelectrical cables and, potentially, they can be used for characterizing the engineering structures for several meters, as there are no limits to the depth which can be allocated the electrodes. The only limitations of the methods are due to the decrease in resolution with the increasing distance between boreholes and to the necessity to have a good contact between the soil and electrodes [29,30].

In this paper, the combined use of minimally or/and non-destructive methods was carried out on the characterization of an analog physical RC model built in the laboratory (Hydrogeosite Laboratory of CNR-IMAA, Marsico Nuovo, PZ, Italy). In detail, a multi-sensor geophysical approach, GPR and ERT methods, was used to define their potentiality and applicability for engineering issues. Therefore, the GPR was carried out on the detection of the structural ‘skeleton’ of a building, where the RC frame of a network of columns and connecting beams are defined. This grid of beams and columns is typically constructed on a concrete foundation and is used to support the building’s floors, roof, walls, cladding, and so on. Several engineering issues are related to RC structures, such as the rebar rusting. This ruins the durability of concrete structures in ways that are difficult to detect and costly to repair. Before addressing this issue, on which our research will pay attention in the next experimental work on the same analog RC model, the first part of the paper wants to focus on a post-build construction phase to check the correct correspondence between the engineering plan and the reinforcements present in the built structure. The second part of the paper depicts the GPR capability on the detection of two different foundation structures, also introducing the ERT approach with electrodes located both on the surface and along small holes. The target of this application concerns the check of the foundations including the geoelectrical behavior of the subsoil.

## 2. Materials and Methods

### 2.1. Analogue Engineering Infrastructure Model: Laboratory Tests

The paper aims to evaluate the contribution of a geophysical multi-sensor approach for the characterization and monitoring of an RC engineering structure, built ad-hoc at the CNR–IMAA Hydrogeosite laboratory of Marsico Nuovo (Basilicata, Italy). The structure was planned and built for performing geophysical experiments in controlled conditions; for this reason, all the constructive details are known to reduce possible uncertainties of the geophysical tests. The RC frame was built with concrete C28/35 reinforced with B450C steel. The engineering design is defined by two columns (C1 and C2) high 170 cm with a square plan of sizes 30 × 30 cm linked on the bottom and the top by two beams B1 and B2 with sizes 360 × 30 × 30 cm, respectively as shown in Figure 1.

The lower beam, linking the two foundations, is placed under a concrete slab thick 10 cm, reinforced with an electro-welded mesh with a mesh size of 25 × 25 cm and a wire size of 8 mm. At the bottom of the two pillars, two different systems of shallow foundations are designed and built: the first one (F1 in Figure 1), a rigid foundation, is a pad footing with a square base of 130 cm, the second one (F2 in Figure 1) is a flexible foundation, constituted by a sloped footing with a square base of 110 cm. Both F1 and F2 are classified as isolated footing (shallow foundations), usually adopted when the ground, at the construction level, has homogeneous satisfactory mechanical resistance for the structure load. The foundations were built on geological sediments characterized by clay and silt and the excavated area was filled with heterogeneous material including gravel and sand. The realization of the two foundation structures is justified by the necessity to identify the geophysical behavior of structures characterized by different sizes, shapes, and reinforcement content that commonly can be found in the RC buildings realized in the last fifty years in Italy. Moreover, the sloped footing is characterized by less use of reinforcements and concrete and it is a typical structure used on old civil structures, on the contrary, the pad one represents the normal standard for civil construction. The frame and the two foundations are realized considering the Italian technical standards (NTC 2018) with the requirements for rigid and deformable soils to have a wider casuistry for the geophysical tests yet performed and to be performed in the future.

Two different diameters are used for the longitudinal rebars of the columns, 12 mm for the central reinforcements and 16 mm for the ones placed externally. Instead, B1 and B2 are reinforced with six rebars of diameters of 12 mm and 14 mm. The transverse reinforcements used for C1 and C2 are constituted by ties with a diameter of 8 mm placed every 7 cm near the nodes and 10 cm far from the nodes. Similarly for the beams, where the stirrups of 8 mm are placed every 6 cm near the nodes or 20 cm far from the nodes. The reinforcements adopted for the foundations are constituted by rebars with a diameter of 16 mm that are placed at variable distances. Details about the reinforcement plan of the structural elements are shown in Figure 2.

### 2.2. GPR Data Acquisition and Processing

GPR surveys were performed with the SIR-3000 System (GSSI Company, Nashua, NH, USA) coupled to three different antennas operating to the frequencies of 400, 900, and 2000 MHz. The selection of the frequency antennas is justified by operational issues and the depth of investigation researched. In detail, thanks to the limited sizes of the 2000 MHz antenna (10 cm × 10 cm), it was possible to analyze the columns and beam (B2) constituting the frame; while the 400 MHz and 900 MHz antennae were exploited to localize and reconstruct the foundation structures placed at a depth not reachable by the first antenna. Indeed, despite the 2000 MHz antenna provides a better resolution for the reconstruction of the reinforcement plan adopted for the frame, the investigation depth reachable is lesser than 30 cm, much less than the one required to detect the structure placed in the subsoil. All the data sets were acquired with the support of an incorporated odometer (survey wheel). The columns and beams were investigated with 250 profiles by 2000 MHz antenna. They were acquired in two directions every 3 cm according to the scheme shown in Figure 3a. The GPR survey with the 400 and 900 MHz antennae was focused on the investigation of the underlying structures in two different steps (before and after the removal of the RC slab). In detail, 36 parallel profiles in two perpendicular directions (24 along the X-axis and 12 along the Y-axis) were acquired with the 400 MHz antenna at a mutual distance of 20 cm; while, with the 900 MHz antenna, 48 profiles were acquired along the X-axis and 24 along the Y-axis at a mutual distance of 10 cm (Figure 3b).

The setting parameters used for the different working frequencies during the data collection phase are reported in Table 1. No gain and frequency filters were used during the acquisitions. GPR data were analyzed with the Reflex-w software [31]. The GPR data were elaborated according to the processing chain showed in Figure 4 with the adoption of few basic operations to safeguard the quality of the data and to avoid unwanted artifacts. The first step consisted of the editing of the data that has permitted to assign the coordinates to each radargram acquired. The start time of each acquisition was corrected to the main bang removing information of no value contained in the radargram. A background removal filter was applied to remove any noise affecting the data due to the clutter effects.

The measured signal was amplified using a linear function of gain to compensate for the intrinsic attenuation of the EM signal and a band-pass frequency filter was implemented to reduce any noise increased by the gain function previously adopted. Indeed, frequency band filters (bandpass butter-worth) to mitigate effects recorded at frequencies too far from the operating one characterizing the single antenna were used. Then, the data were migrated with the use of the Kirchhoff algorithm based on an estimated EM waves velocity equal to 0.11 mns^−1^ (dielectric permittivity of 7.5), approximately. No substantial changes have been noted about the EM velocity propagation for the different structural elements investigated. The velocity values are obtained by analyzing the diffraction hyperbolas generated by the internal iron bars to the laboratory structure and their geometry. The time it takes for the waves to travel to their vertex is a function of the depth of the object, as well as the velocity of the waves entering the material [32]. In the final step, a 3D representation of the entire investigated laboratory building structure was created.

### 2.3. ERT Data-Acquisition and Data-Processing

The geoelectrical applications were carried out in two steps. During the first one, the resistivity data were acquired with a cross-hole approach with electrodes located in the boreholes. The CH-ERTs were carried out using a prototypal system designed and realized ad hoc including two multichannel PVC pipes with 15 small cylindrical stainless-steel electrodes (AISI 316) installed around. The electrodes were installed 10 cm between 0 and 1.5 m depth and each one connected with the georesistivimeter with a single-ended electrical cable. The instrumented PVC pipes were installed in two holes, drilled into the slab and soil close to the foundation F2 (Figure 5). To reduce the electrical contact resistance between the electrodes and the soils, the holes were filled with a mixture of water and bentonite before the starting of the acquisition and electrodes were wet by an electrically conductive gel. The two wells were arranged according to the scheme shown in Figure 5. Considering the F2 foundation maximum width, the distance between boreholes A-B was set to 150 cm. Cross-hole electrical resistivity data were acquired using an electrode configuration (well-well quadrupole array), using the energizing electrodes placed in A and the potential electrodes in B.

The first goal was obtaining a high-resolution section of the electrical resistivity variation between the two boreholes to obtain information at greater depth about the interaction between the subsoil and the engineering structure and, possibly, to define the geometry of the foundations.

During the second step, the surface RC slab was removed with the help of a mini-excavator that has provided to destroy and dismantle this element without minimally compromising the other elements of the structures. At this stage, eight 2D ERTs were acquired on the ground surface as showed in Figure 5 with conventional electrodes fixed on the ground. We decided to remove the slab as the acquisitions performed with steel plate electrodes placed directly on the floor did not provide good results. In detail, 29 electrodes spaced 20 cm, and two multichannel cables were used for each line. The ERTs were spaced 15 cm for investigating both the area above the foundations and the backfill.

The ERTs were acquired using the SYSCAL Pro Switch resistivity instrument (Iris Company) powered by an external 24 V battery and 96 channels. A Dipole-Dipole (DD) configuration was adopted for the surface investigations. To reduce the uncertainties and improve the data quality, the reciprocal dataset (obtained by swapping current and potential electrodes) was acquired for the necessary measurements error assessment. To obtain the true electrical resistivity distribution, the apparent resistivity values of all the data set were inverted using the ResIPy software [33].

ResIPy was developed to provide a more intuitive, user-friendly, approach to invert geoelectrical data, using an open-source graphical user interface (GUI) and a Python application programming interface (API). It utilizes the R2/R3t inversion codes for apparent electrical resistivity data inversion based on the widely used Occam’s approach of deGroot Hedlin and Constable [34]. The earth is discretized into a series of parameter blocks, each containing one or more elements. The Occam’s inversion finds the smoothest 2-D model for which the Chi-squared statistic equals an a priori value [35]. In detail, the electrical resistivity data were processed according to the processing chain shown in Figure 6. Apparent resistivity data are imported in the software and using the reciprocal measurements an error model analysis was defined. The next step consists of the generation of a triangular or tetrahedral mesh for 2D or 3D inversion. The mesh and the data are sent to the inversion pipeline and the resulting inverted section is produced with a diagnostic pseudo section of the normalized error of the inversion. The surface and cross-hole apparent resistivity data set was inverted by using a triangular mesh with about 5300 and 11,000 elements, respectively. Moreover, according to Rücker and Günther [36], borehole electrodes were treated as ideal point sources (valid for length/spacing ≤0.2).

Finally, a 3D inversion from all the 2D ERT profiles disposed above F1 and F2 (Figure 5) was carried out. In detail, around 2800 apparent resistivity values were inverting using a mesh with about 40,000 tetrahedral cells, a mixed boundary condition (Dirichlet and Neumann), and a starting homogeneous apparent resistivity of 100 Ωm. The noise was estimated by evaluating reciprocal measurements. In detail, a reciprocal error threshold of 10% was used. 2D and pseudo-3D inversion converged in at most 10 iterations. In all cases, the final model residual error (as in Figure 6f) is between ±3% [35].

## 3. Results

### 3.1. 2D and 3D Reconstruction of the Building’s Elevation Reinforcements

Figure 7 highlights some examples of radargrams collected in correspondence of the upper beam (B2) and the two columns (C1 and C2). Despite the impossibility to use an automatic system of acquisition, 2000 MHz data permit to accurately reconstruct the disposition of the longitudinal and transverse reinforcements (stirrups and tiles, respectively).

The reflections due to the presence of the rebars are observable at depths ranging between 3 and 4 cm according to the design specifications. The most reflective areas are collected in correspondence with the structural nodes, where the simultaneous presence of rebars, stirrups, and tiles generates a highly reflective area in the RC structure. However, the reflections due to the rebars placed on the opposite site of the frame are hardly detectable as shown by the slices placed at the constant depths greater than 20 cm. It is also clear the presence of the RC node confinements where the transversal rebars are placed at a lesser distance from each other, as commonly used to increase the ductility of the structures in seismic areas (green curly brackets in Figure 7b–d).

GPR combined with the limited distance adopted for the data acquisition (i.e., radargrams manually acquired every 3 cm) allowed to three-dimensionally reconstruct the reinforcements of the frame. The depth slices plotted in Figure 8 reveal the presence of the longitudinal and transverse rebars starting from the first 3 cm investigated up to 5 cm, (Figure 8b,c), approximately; this is in accord with the design requirements that indicate a concrete cover of 3 cm. As expected, these reflections disappear at depths greater than 5 cm, where only the concrete is present; indeed, at the depth of 10 cm, the reflections could be associated only to the presence of inhomogeneities generally due to the materials constituting the concrete (Figure 8d). The last slice showed in Figure 8e identifies some reflections supposedly due to the presence of the rebars placed at the opposite site where investigations were performed.

As shown in Figure 9, reflections occurring in the beam and the columns in correspondence of the nodes A and A′ are partially matching with the reinforcement plan. Some discrepancies between the GPR image and the design are imputable to constructive defects made to the workers. For example, the distance required by the project for the column tiles is not respected, as showed by the slices that do not provide the designed variation from the transverse reinforcement between the node and the middle of the column. However, the number of rebars and the distance between them agree with the design requirements. Finally, the rebars used to ensure the structural continuity of the reinforcements coming from the foundations, nodes B and B′ in Figure 9, are not perfectly detectable except by an increase of the reflectivity characterizing the slices in these areas. Indeed, a more reflective area characterizes the bottom of the columns for a length of about 60 cm. This is in good agreement with the “desired” length of the rebar overlap fixed to 70 cm.

### 3.2. Characterization of the Foundation Structures with GPR Data

Figure 10 shows the radargrams acquired with antennas at 400 MHz and 900 MHz frequency (Figure 10b,c). They highlight the presence of several reflections associable to the upper RC slab (yellow dotted lines), placed above the foundation beam and the two different foundation structures, respectively. As expected, it is easily detectable the reinforcements occurring in the slab above which the data are collected (welded mesh). The higher resolution of the 900 MHz antennae has provided the possibility to significantly improve the reconstruction of the mesh. The beam that links at the top of the two foundations is also easily detectable using both datasets. The reflections placed under the first 40 cm and 30 cm, for the investigations achieved in the presence (Figure 8b,c) or not (Figure 8d,e) of the RC slab, respectively, are clearly associable to the presence of the foundation structures (highlighted with red polygons in the radargrams). However, other reflections are visible between the two foundations imputable to the heterogeneous materials used for filling the excavated area after the realization of the structures.

Figure 11 shows the horizontal slices extracted by the 3D volume at constant depths for evaluating the capability of the GPR for the localization and characterization of the two structures. In particular, the data collected at the frequency of 400 MHz allow us to identify some reflective areas associable with the presence of the different structures placed in the subsoil (Figure 11a). The shallower results (Figure 11b) provide information about the presence of the reinforced slab placed at the bottom of the columns; while the other slices allow identifying the bottom beam B2 (Figure 11b) and the F1 and F2 foundations (Figure 11c–g). The slices placed at the depth ranging between 0.3 m and 0.9 m intercept both the foundations, while the deeper ones show only the rigid foundation.

It is evident the presence of several reflections reduces the capability of the GPR for the localization and characterization of the underground structures; this is further complicated for the presence of a square sewer well placed near the rigid foundation. The strong reflections produced by this object do not provide the detection of the quadrangular shape of the deeper structure. Further, the heterogeneities due to the bad leveling of the subsoil after the realization of the structure have produced a lot of spurious reflections which greatly complicate the identification of the structures placed in the subsoil.

### 3.3. Characterization of the Foundation Structures with ERT Methods

ERT profiles L2 and L6 are respectively located directly on the backfill (Figure 12a); L2 outside the foundation (Figure 12b) and L6 above the two foundations system (Figure 12c). The electrical resistivity values range between 1 and 10,000 Ωm. This high resistivity range is due to the very variable nature of the backfill used for filling the excavated area where the two foundations are realized. The ERTs L2 and L6 (Figure 12b,c) show a general resistive behavior (>200 Ωm) associated with the backfill, while quite lower electrical resistivity zones (<50 Ωm) are present in correspondence of both foundations. However, the surface ERT survey has a limited depth of investigation (up to 80 cm), therefore it is not possible to observe all the foundation zone.

To have a complete observation of the subsoil and the buried investigated structures (F1 and F2), also giving some details about their shapes and sizes, a 3D elaboration approach was introduced. Figure 13 shows a slicer view of the 3D inversion of the ERT profiles at different depths (0.2 m, 0.4 m, 0.6 m, and 0.8 m). The slices highlight the electrical resistivity behavior described before. As expected, the presence of the two foundations strongly influences the electrical resistivity behavior of the subsoil (Figure 13). In detail, the low resistivity anomaly (<50 Ωm) on the left is due to the presence of the foundation F1 while the right low resistivity anomaly detects the F2 foundation. Even if the maximum investigation depth is still 80 cm, which is less than the dimension of the foundations. The 3D ERT image highlights well the correlation between the engineering structures and the backfill material. The strong contrast permits us to depict the observable part of the foundations.

Figure 14 shows the cross-section image of the CH-ERT inversion. CH-ERT was acquired with the presence of the RC slab thick 0.1 m in correspondence with foundation F2, reaching the investigation depth of about 1.5 m. CH-ERT is characterized by a high resistive layer (>500 Ωm) down to 0.4 m depth and quite low resistivity values (<50 Ωm) in correspondence with the foundation. Small lateral resistivity contrasts between the backfill and the foundations characterize the CH-ERT results. Some important aspects are well highlighted using the electrical resistivity tomography CH-ERT, an example is the depth of investigation that is the same as the borehole depth (140 cm).

## 4. Discussion

Non and low-invasive assessment of the safe conditions of reinforced concrete structures is an essential component of scientific research aiming to provide useful information to support the work of engineers during the design and realization phases of maintenance projects. To characterize engineering structures through a laboratory test, this paper has discussed the applications of GPR and introduced the 3D surface ERTs, moreover, the ad hoc CH-ERT acquisition test was improved. In detail, the first part of the paper highlights the GPR capability on the observation of the skeleton of a building, which is important during RC frame check (i.e., after an earthquake event to check the status of the rebars). The second part introduces a comparison between the classical GPR methodology and a new geophysical observation approach for engineering buried structures (CH-ERT method), to define the possible observation improvements.

The cooperative use of different sensors demonstrates that the two methodologies can effectively support more conventional non-destructive techniques (acoustic and thermal methods) for the vulnerability analyses required within the engineering field.

Indeed, GPR results collected at the frequency of 2000 MHz provided a lot of information about the technical details of the structure; in particular, the characterization of the upper beam and the two columns, allowed to identify the distance between the stirrups and tiles and to localize their position. Moreover, by comparing the obtained results with the design drawings some useful consideration can be achieved.

As regarding the reinforcements presents in the beam/column nodes, GPR radargrams have allowed us to clearly distinguish the presence of the tiles, stirrups, and longitudinal steels placed in according to the reinforcement design; some discrepancies, due to the technical errors made during the realization of the structure are detected near the nodes as highlighted in Figure 15.

The data obtained at the lower frequencies were addressed to identify and reconstruct the structures placed under the visible structure. Results have permitted to identify some reflections imputable to the two foundations although the high heterogeneity of the subsoil. In particular, the best resolution, obviously obtained at the frequency of 900 MHz, allowed to detect and reconstruct the structures with more detail up to a depth of 1 m, approximately. However, data collected at the 400 MHz frequency has been provided to identify the laying surface of the deeper structure.

However, it is worth noting that, regarding the foundation detection and characterization topic, the interpretation of the data is far from being obvious, as shown in Figure 8. Indeed, four main causes have reduced the capability of the GPR to detect the structures: (i) presence of reinforcements into the slab, (ii) presence of the foundation beam, (iii) presence of some voids in the subsoil due to inadequate soil compaction before the realization of the structure, (iiii) a not homogeneous soil used for the filling of the excavated areas. Therefore, it is evident the importance to have a priori information about the engineering structures to support their characterization during the usual maintenance and restoration operations.

Really interesting results are obtained with the ERT method. It is based on the introduction of DC current in the subsoil and even if this approach is commonly used for different kinds of geological issues, on the contrary, the CH-ERT method, mostly used for hydrogeological application, is still considered a non-conventional method for its large performance variability and, therefore, several improvements are necessary to apply it in the engineering field. Cross-hole and surface electrical resistivity imaging underline the presence of conductive anomalies (<50 Ωm) in correspondence of foundations structures (Figure 14). However, cross-hole ERT, despite its greater depth of investigation, did not have the resolution such as to be able to identify the lateral passage between backfilling and foundation structures. In particular, the sensitivity analysis for cross-hole measurements indicates that boreholes A-B are probably too far apart from each other and therefore the method could not be ideal for this type of foundation.

The ERT acquired on the surface has been permitted to three-dimensionally reconstruct the anomalies characterizing the subsoil allowing us to clearly identify some electrically more conductive nuclei due to the RC foundations.

Despite the general electrically resistive behavior of the concrete [36,37], showed laboratory results where the reinforcement greatly influenced the resistivity response of concrete causing the recording of relatively low electrical resistivity values. In this work, both 2D and 3D analysis of a real foundation structure confirm the laboratory results to a real observation scale and in a background contest very close to a real one.

Finally, the overlapping between design, radargram, and ERT (Figure 16) highlights how a first level of the integration of the collected data can effectively support the detection of the two foundation structures limiting the uncertainties characterizing the reading of the results obtained with the employment of only one method. As showed in Figure 16a, where the rigid foundation structure was intercepted, the CH-ERT has detected a conductive anomaly and GPR has identified multiple reflections. Still, in presence of the two foundation structures intercepted by the geophysical surface analyses (Figure 16b,c), the agreement between the e-m reflections and the electrical anomalies is clear and allows to define positions and shapes of the intercepted structures

## 5. Conclusions and Future Perspectives

The strong urbanization taking place on a global scale requires the development of new methodologies that make a strong contribution to the activities of characterization and monitoring of interventions in the engineering field. In this scenario, in addition to the classic diagnostic methodologies, geophysics can provide a valuable contribution. With this work, we have analyzed how some geophysical investigations, which have a non-invasive or minimally invasive character, can provide useful information about the engineering structures realized in RC. Some issues are well detected in this kind of structure, such as the rebar rusting. This phenomenon strongly affects RC structures; therefore it is necessary to focalize scientific work on that. Indeed, our work will move in this direction and the analog engineering model in the laboratory will help the community to make it. This paper is the starting point of our project, where a checking design phase and a geophysical method evaluation were introduced.

GPR measurements conducted in a non-invasive way have allowed us to accurately define the most important details for the evaluation of the state of conservation for RC structures; indeed 2 GHz results have provided useful information about the analyses of the concrete cover thickness, a diameter of the rebars and nodes reinforcements. This information is essential for a correct evaluation of the resistance and resilience of the common RC structures, in particular, for the ones realized in seismic zones where structural details are fundamental for preventing severe damages.

Further, GPR data allowed to verify the agreement between the design requirements and the construction, highlighting also a constructive error relative to a poor reinforcement of the nodes as discussed in Section 3.1.

The measurements realized at the frequency of 400 and 900 MHz antennae have identified, albeit, with difficulty, the foundation system adopted for the construction. This topic is of considerable importance because it allows us to investigate the zone of subsoil, which we can define as the “*infrastructure critical zone*”, where the most important issues for engineering structures can be found, and for this reason deserves greater attention especially when external events can cause cracks in the structures that are not easily visible. The investigations carried out on the reinforced concrete slab gave the possibility to observe the electro-welded mesh present, while the delineation of the underlying foundation systems was more difficult. With the help of the data design, it was possible to identify the main reflections inherent to the presence of the foundations. The investigations were repeated by removing the reinforced concrete slab and making the antenna directly in contact with the ground. The research carried out on the ground showed a significant improvement in the signal, but this result does not solve the resolution problem encounter for the detection of the foundation geometries.

To provide useful information able to support the GPR results about the presence and characterization of the foundation structures, several electrical resistivity tomographies, realized with electrodes placed in boreholes and on the surface (after the removal of the RC slab) were performed. Cross-hole electrical resistivity (CH-ERT) was used with the electrodes placed inside two mini-holes. The results obtained made it possible to identify the presence of the foundation system interposed between the two wells in the central part as a body with low electrical resistivity. From the section of a variation of the electrical resistivity, it was possible to observe an electro-layer with higher resistivity in the upper part due to the presence of a very heterogeneous and poorly compacted filling soil.

Following the removal of the reinforced concrete surface slab, surface ERTs were performed. The results are in agreement with the ones obtained with CH-ERT.

The comparison and integration of GPR and ERT results can effectively support the detection and characterization of the structures; however, they require further laboratory experiments to maximize the information quality. The data obtained, if not carefully evaluated by qualified personnel, can be difficult and not immediate to interpret. Experimenting on a building analog built in the laboratory was found to be of fundamental importance in establishing the potential and limitations of the geophysical survey techniques adopted. This work, indeed, has highlighted how the use of non-invasive geophysical investigation techniques for engineering characterization show significant potential to the obtaining of clear and concrete information on the structure of existing buildings to plan restoration, renovation, and interventions more substantial and invasive of expansion or concerning the safety of existing urban building structures. Future activities will involve the realization of a more homogeneous filling for the subsoil, to reduce the uncertainties affecting the recent results and, further, offer an optimal test site for studies based on the use of geophysical methodologies (electric, electromagnetic and seismic) for RC corrosion analyses. Further, the use of minimally invasive technologies is a valid tool for effectively supporting the development of new strategies and programs for the planning of urban centers to make them resilient to natural disasters and to increase energy and environmental sustainability [38]. Indeed, the advancements required for improving the knowledge of heterogeneous subsoil of urban and industrialized sites, often characterized by the absence of information about the constructions and realization of the infrastructures and structures represent a challenging topic for the urban geophysics only solvable with the integration and cooperative use of different methodologies acting at different scale and resolution. This is particularly felt for the study of those structural elements not easily investigable such as in the case of the foundation structures constituting what can be rightly called an infrastructural critical zone, including those levels of subsoil that are more or less anthropized [23,39,40,41,42].

## Figures and Tables

**Figure 1 sensors-21-05565-f001:**
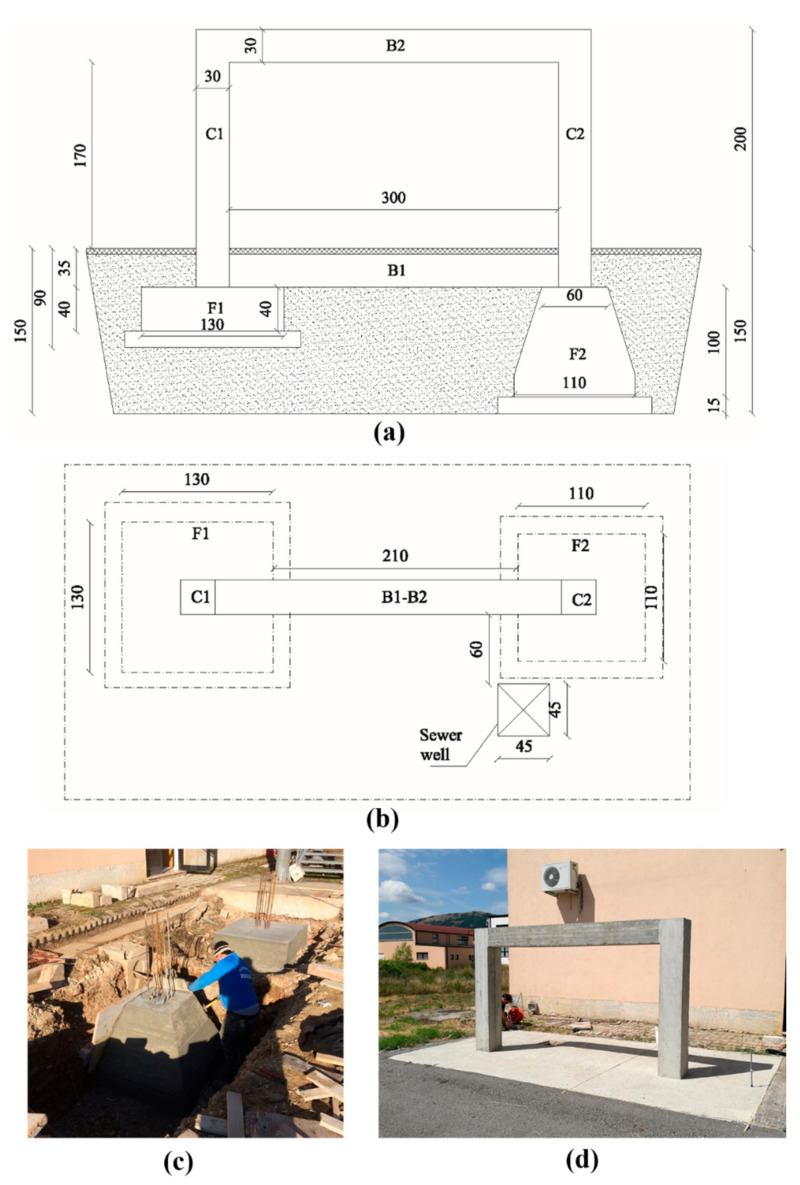
Sketch of the RC frame realized at the Hydrogeosite Laboratory (**a**,**b**): the structure is constituted by two columns (C1 and C2) and two beams (B1 and B2). Two different types of foundations are realized (F1 and F2). Photos of an intermediate constructive phase (**c**) and the final construction (**d**).

**Figure 2 sensors-21-05565-f002:**
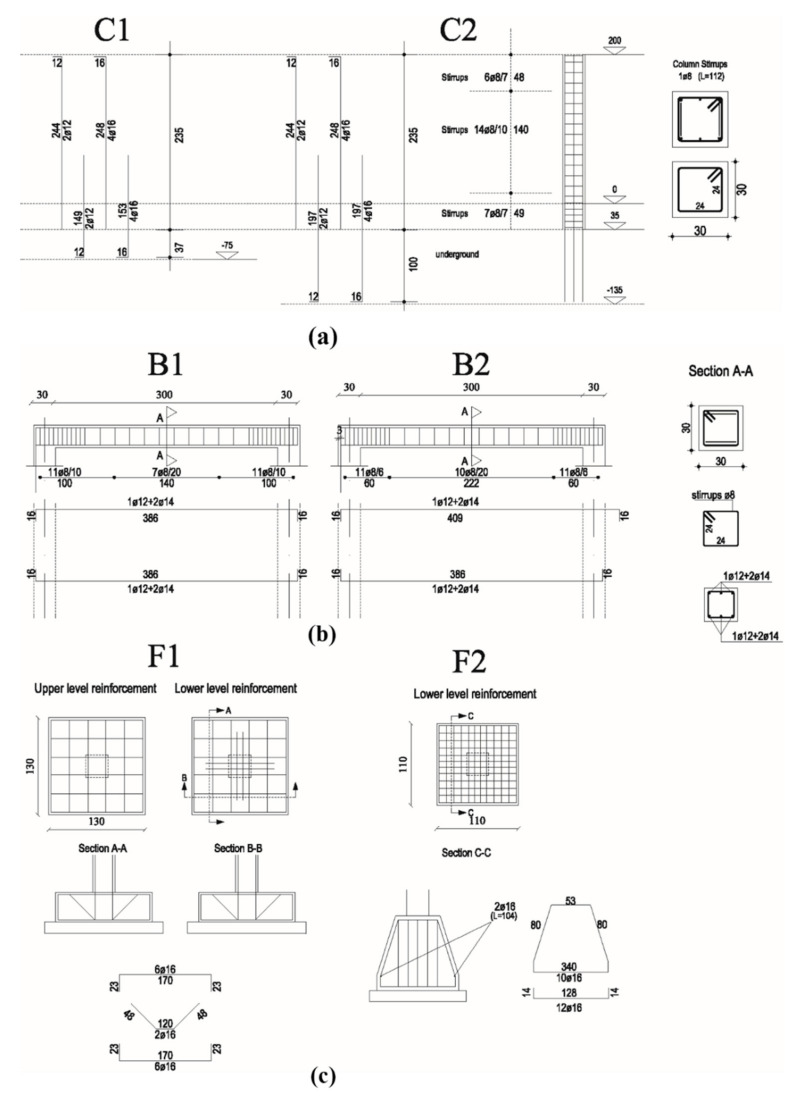
Reinforcement plans for the structural elements constituting the RC frame: reinforcements designed for the columns (**a**), beams (**b**), and foundations (**c**). All distance units are given in cm, whereas the diameters of reinforcing bars are expressed in mm.

**Figure 3 sensors-21-05565-f003:**
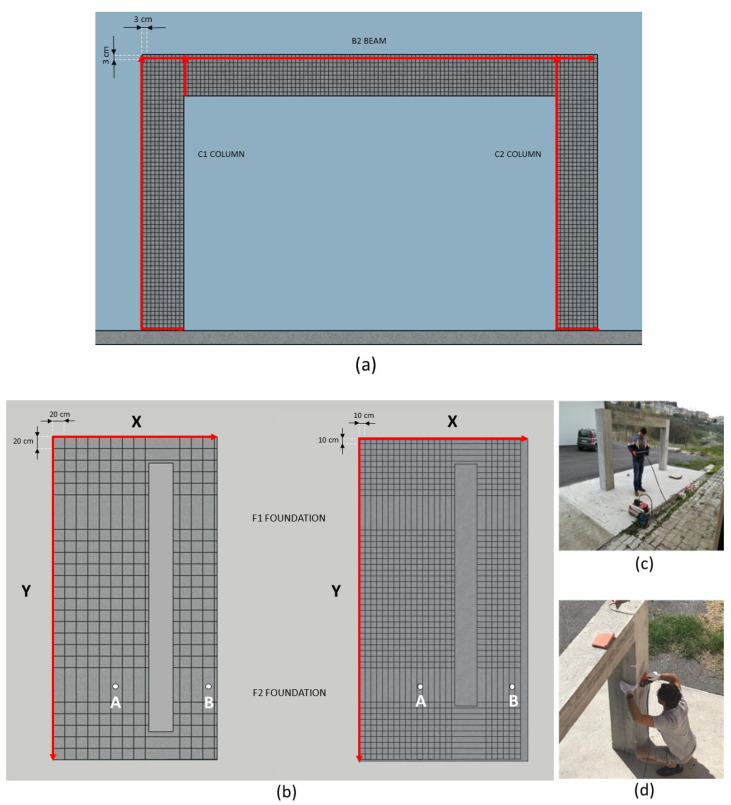
Acquisition scheme adopted for the 2000 (**a**), 400, and 900 (**b**) MHz antennae. In (**b**) are indicated the position of the two boreholes A and B, used for the CH-ERT, close to the foundation F2. In (**c**,**d**) are showed some phases of the GPR data collection on the floor and the column, respectively.

**Figure 4 sensors-21-05565-f004:**
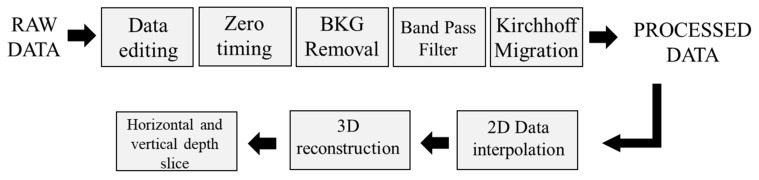
Processing Chain adopted for GPR data processing.

**Figure 5 sensors-21-05565-f005:**
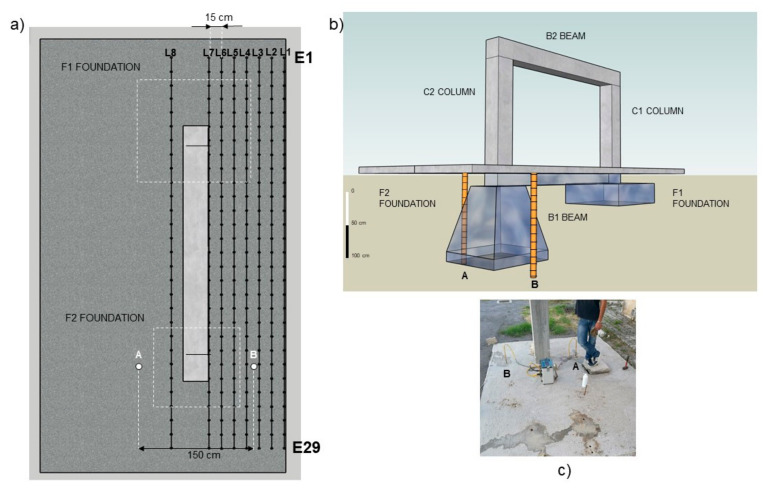
Localization of the surface ERTs was collected in correspondence of the floor (**a**); 29 electrodes were used, the first one placed in correspondence of E1 and the last one of E29. As you can see only one side of the foundation structure was investigated. Two boreholes are realized near the rigid foundation (F1) where the two geoelectric cables were installed (**b**). The measurements were performed with the Syscal Pro 96 Channel georesistivimeter and the contact resistances are reduced with the use of conductive gel (**c**).

**Figure 6 sensors-21-05565-f006:**
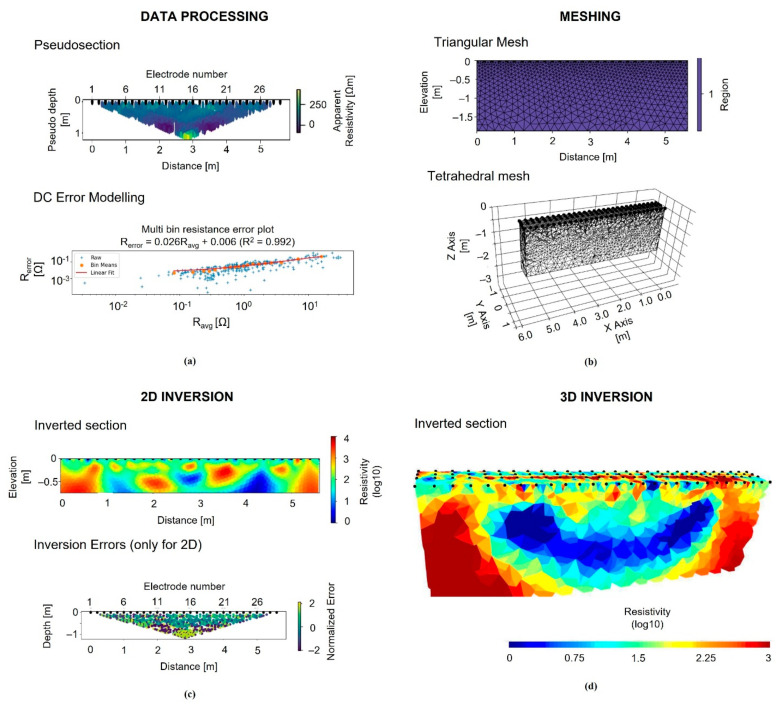
Processing flow adopted for electrical resistivity data processing. The first step is the importation and analysis of the apparent resistivity data (**a**). The second one consists of the generation of a mesh (**b**). The last one, the inversion process, is performed with the use of 2D (**c**) and 3D (**d**) collected data.

**Figure 7 sensors-21-05565-f007:**
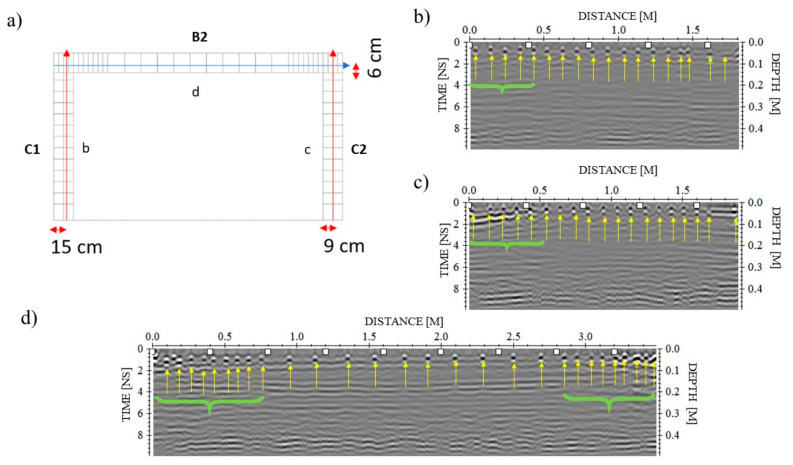
The sketch of the rebar frame in the two columns and beam (**a**). Radargrams acquired in correspondence of the two columns C1 and C2 (**b**,**c**) and the upper beam B2 (**d**) with an indication of the rebars (yellow arrows). The curly brackets in green highlight the distances where the transverse rebars are placed at a reduced distance.

**Figure 8 sensors-21-05565-f008:**
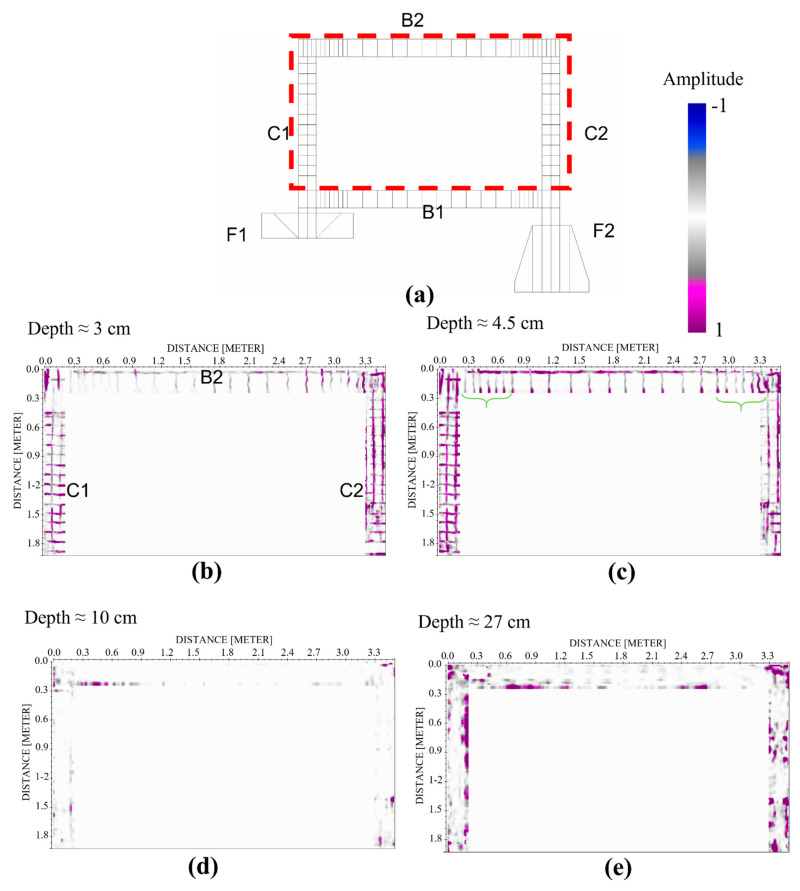
Reinforcement plan of the columns (C1 and C2) and upper beam (B2) obtained with the GPR 2000 MHz measurements: the reinforcement design (**a**) used for the realization of the structure; the depth slices corresponding to the depths of 3 cm and 4.5 cm allow to identify the longitudinal and transverse rebars (**b**,**c**); on the contrary, at the depth of 10 cm no reflections are detected (**d**). The rebars placed near the opposite sides of the structure are not clearly detectable (**e**).

**Figure 9 sensors-21-05565-f009:**
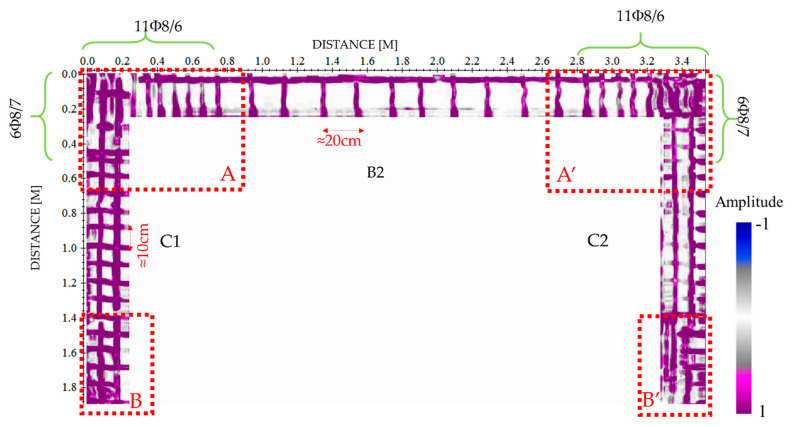
Identification of the constructive details used for the construction of the frame with the GPR analyses performed at the frequency of 2000 MHz (depth slice collected at the depth of 4.5 cm) where the reflections induced by the reinforcements present in the structural nodes (red dotted rectangles) are shown.

**Figure 10 sensors-21-05565-f010:**
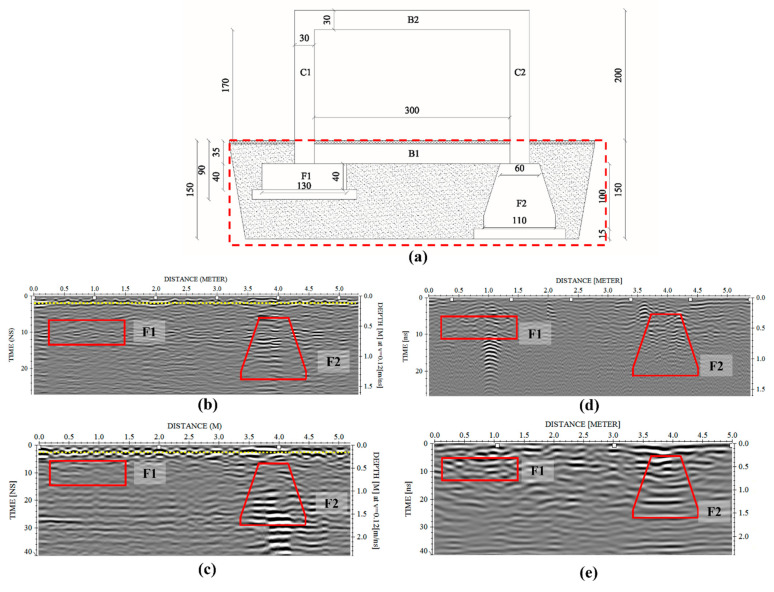
Radargrams were acquired in correspondence of the foundation structures F1 and F2 (**a**) at the frequencies of 900 MHz (**b**–**d**) and 400 MHz (**c**–**e**). The acquisitions are carried out directly on the ground (**d**,**e**) and the reinforced slab (**b**,**c**). The red polygons show the positions of the two structures where the most remarkable reflections are detected.

**Figure 11 sensors-21-05565-f011:**
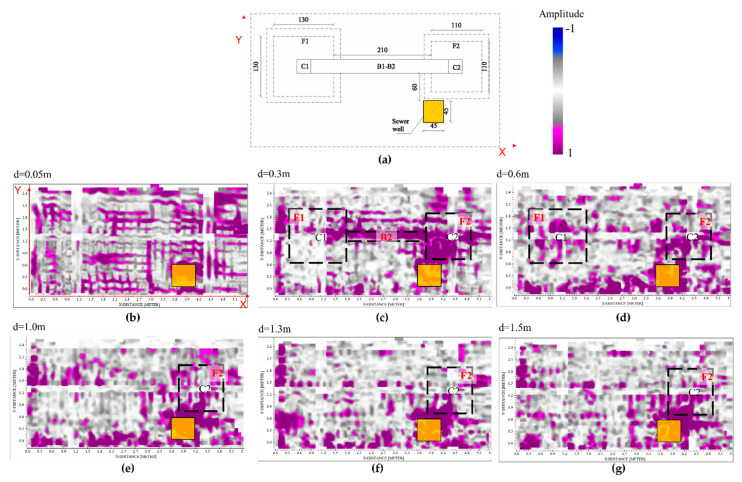
Horizontal slices extracted by the 3D models (400 MHz datasets) for foundation detection and characterization. The data were recorded on the RC slab of the structure according to two perpendicular directions (**a**); at the depth of 0.1 m only the iron mesh was identified (**b**), at the depths ranging between 0.3 m and 0.6 m, some reflections associable to the presence of the two foundations and bottom beam are detectable (**c**,**d**). The slices placed at greater depths allow one to identify reflections imputable to the presence of the deeper foundation, the rigid one (**e**–**g**).

**Figure 12 sensors-21-05565-f012:**
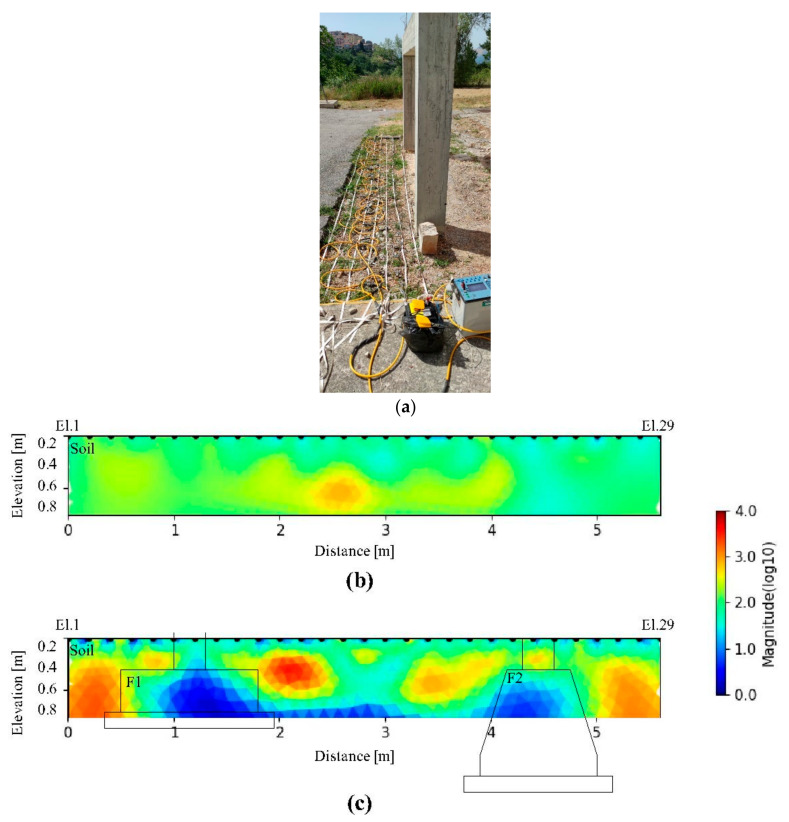
2D ERT profiles on the backfill (**a**). L2 (**b**) was acquired where no structures are present, while L6 (**c**) intercepts part of the foundations F1 and F2. L2 and L6 profiles were carried out directly on the ground. The black polygons show the positions of the two buried structures where relatively low electrical resistivity values are depicted.

**Figure 13 sensors-21-05565-f013:**
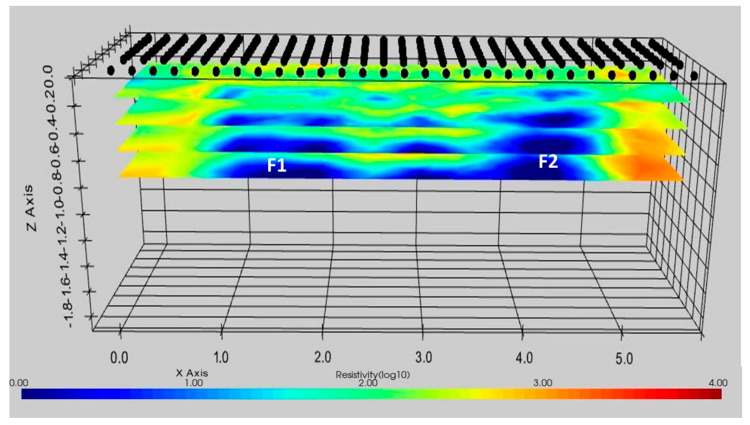
3D ERT from the inversion of all 2D lines. The two foundations (F1 and F2) were localized.

**Figure 14 sensors-21-05565-f014:**
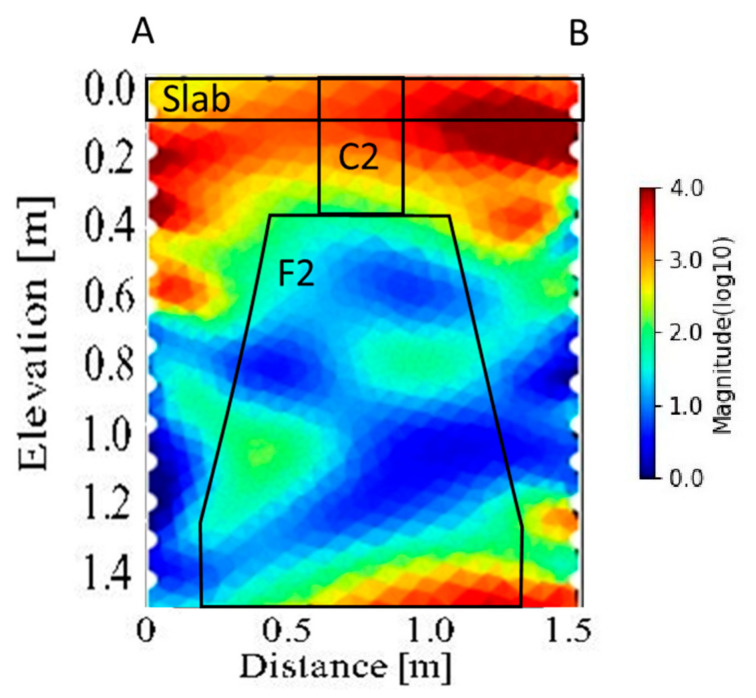
CH-ERT acquired between the foundation F2.

**Figure 15 sensors-21-05565-f015:**
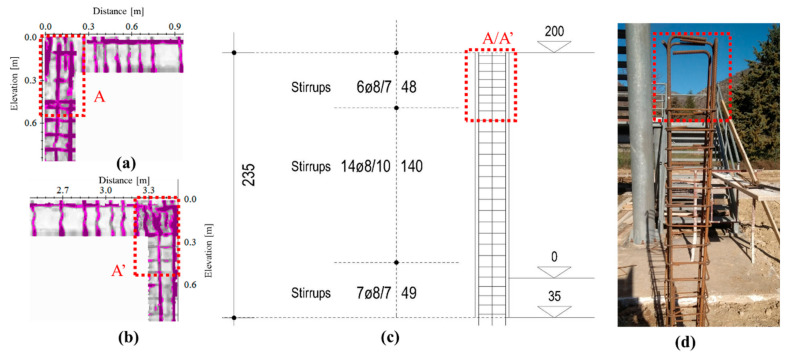
Comparison between the slices obtained at the depth of 5 cm, approximately (**a**,**b**), and the planned reinforcement of the columns (**c**). Results confirm the discrepancies between the realized structure and the designed one; indeed, the stirrups are totally based on the correspondence of the structural nodes A and A’. This is also validated by an internal photo made during the constructive phase of one of the columns (**d**).

**Figure 16 sensors-21-05565-f016:**
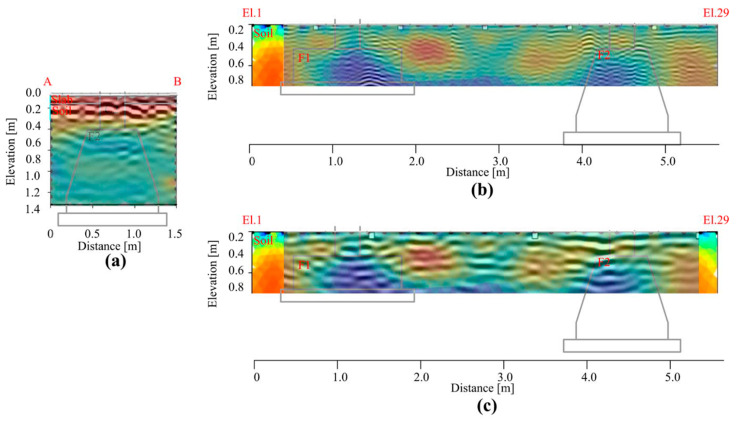
Overlay of the GPR and ERT data acquired in the borehole (**a**) and on the surface (**b**,**c**) with the use of radargrams acquired at the frequencies of 900 and 400 MHz, respectively. The conductive anomalies present in the ERTs correspond to the two more reflective areas of the radargram.

**Table 1 sensors-21-05565-t001:** Data acquisition settings of the three used antennae on the RC structures.

Antenna Frequency	Trace Increment (Trace/cm)	Time Increment (ns)	N. Sample	Time Window (ns)
400 MHz	0.5	0.136	512	70
900 MHz	1	0.058	512	30
2000 MHz	3	0.023	512	12

## Data Availability

The data presented in this study are available on request from the corresponding author.

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
