# Peer review of "Multi-Sensors Geophysical Monitoring for Reinforced Concrete Engineering Structures: A Laboratory Test"

_sensors, 2021, doi:10.3390/s21165565_

Round 1

Reviewer 1 Report

  1. How to select the working frequency of GPR and what principles are based on.? Such as 400Khz 400, 900 and 2000 MHz.
  2. What is the setting principle of filter to remove noise affecting data? How to consider?
  3. How to remove the surface RC slab? Do you mean to destroy and dismantle the concrete slab?
  4. What is the theory of 2D or 3D data inversion? Could you elaborate it?

Author Response

Reviewer No 1 comments:

Authors: Dear Reviewer n.1, we thank you very much for your review and we appreciated your work a lot. We have completed our revision, considering all your comments and suggestions, which improved the paper. Moreover, we tried to improve the editing of English language.

  1. How to select the working frequency of GPR and what principles are based on.? Such as 400Khz 400, 900 and 2000 MHz.

Authors: We thank the reviewer for the comment. The selection of the frequency antennas is justified by operational issues and depth of investigation researched. In detail, thanks to the limited sizes of the 2000 MHz antenna (10cmx10cm), it was possible analyze the columns and beam (B2) constituting the frame; while the 400 MHz and 900 MHz antennae were exploited to localize and reconstruct the foundation structures placed at depth not reachable by the first antenna.

  1. What is the setting principle of filter to remove noise affecting data? How to consider?

Authors: GPR data were analyzed with the Reflex-w software and the GPR data were elaborated according to the processing chain showed in figure 4. In detail, a background removal filter was applied to remove any noise affecting the data due the clutter effects and a frequency band filters (bandpass butter-worth) to mitigate effects recorded at frequencies too far from the operating one characterizing the single antenna were used.

  1. How to remove the surface RC slab? Do you mean to destroy and dismantle the concrete slab?

Authors: During the second step, to carry out the ERT with electrode on the ground, the surface RC slab was removed with the help of a mini-excavator that has pro-vided to destroy and dismantle this element without minimally compromise the other elements of the structures.

  1. What is the theory of 2D or 3D data inversion? Could you elaborate it?

Authors: All the ERT data were inverted using the Python version of the R2/R3t inversion codes for apparent electrical resistivity data inversion developed by A.Binley. It is based on the Occam’s approach inverse algorithm. The earth is discretized into a series of parameter blocks, each containing one or more elements. The Occam’s inversion finds the smoothest 2-D model for which the Chi-squared statistic equals an a priori value. We think it is not necessary to described in detail the inversion theory of the used algorithm because it is detailed described in the A. Binley papers that we have indicated in the text.

Reviewer 2 Report

Applying ER methods in civil engineering is recognized as a useful investigation tool but still remaining a complementary. The paper is clearly structured and wisely conducted. The only concern here is the scientific contribution. All the methods used in the study are widely and commonly applied. The authors are encouraged to expose what's really unique about the research. There is a lack of clear distinguishing between method application is structural and geotechnical engineering. The result is that the study concerns both aspects (narrow perspective in term of geotech. eng.) but non of them are fully covered. When it comes to methodology of the physical model is not fully explained. Since investigating the working or technical condition of the structure depend on the ground condition, then more information on soil parameters should be provided. When it comes to foundation design and dimensions there is now information on limit states that governs the foundation type and structure. This would make the study more complete and reliable. These are the major comments, all other remarks could be followed in the copy revised copy. To name just a few: Please do not use abbreviation in the title, More specific outcomes should be provided in the abstract, More background on the objectives in the intro. need to be presented, The quality of marked figures should be enhanced, Legend explaining the colours should be added to all the figures to make it easier to understand. fig 17 The fig doesn't show much could you please explain in more details.

Author Response

Reviewer No 2 comments:

Applying ER methods in civil engineering is recognized as a useful investigation tool but still remaining a complementary. The paper is clearly structured and wisely conducted.

Authors: Dear Reviewer n.2, we thank you very much for your review and we appreciated your work a lot. We have completed our revision, considering all your comments and suggestions, which improved the paper. Moreover, thank you very much for your positive words on our work. Finally, we have improved the English in the text.

The only concern here is the scientific contribution. All the methods used in the study are widely and commonly applied. The authors are encouraged to expose what's really unique about the research.

Authors: We thank the reviewer for the inspiring comments. We addressed the new version on the reviewer encouragements, highlighting better our aims.

There is a lack of clear distinguishing between method application is structural and geotechnical engineering. The result is that the study concerns both aspects (narrow perspective in term of geotech. eng.) but non of them are fully covered. When it comes to methodology of the physical model is not fully explained. Since investigating the working or technical condition of the structure depend on the ground condition, then more information on soil parameters should be provided. When it comes to foundation design and dimensions there is now information on limit states that governs the foundation type and structure. This would make the study more complete and reliable.

Authors: We agree with the reviewer, the structural and geotechnical engineering aspects are both important even if we used an analogue physical model. This analogue model of foundations and frame were built considering the Italian technical standards (NTC 2018) considering the requirements for rigid and deformable soils to have a wider casuistry for the geophysical tests yet performed and to be performed in the future. The foundations were built on geological sediments characterized by clay and silt and the excavated area was filled by heterogeneous material including gravel and sand.

These are the major comments, all other remarks could be followed in the copy revised copy.

Authors: We addressed all the indicated suggestions.

Please do not use abbreviation in the title

Authors: We have modified the title.

 More specific outcomes should be provided in the abstract,

Authors: We improved the abstract.

More background on the objectives in the intro. need to be presented

Authors: We defined our present and future aims with our research activities. The paper is the starting point of our research project.

The quality of marked figures should be enhanced, Legend explaining the colours should be added to all the figures to make it easier to understand.

Authors: We improved them and we added the colour legend on them

fig 17 The fig doesn't show much could you please explain in more details.

Authors: We modified it and we merged the figure 16 and 17, explaining them in more details, in order to be more simple for the future readers